# Physician exhaustion and work engagement during the COVID-19 pandemic: A longitudinal survey into the role of resources and support interventions

**Lara Solms** [1,2]*, **Annelies E. M. van Vianen** [1], **Jessie Koen** [1,3], **Kees-Jan Kan** [4], **Matthijs de Hoog** [2], **Anne P. J. de Pagter** [2,5], **on behalf of the Improve Research Network** [¶]

1 Department of Work and Organizational Psychology, University of Amsterdam, Amsterdam, The Netherlands, 2 Department of Pediatrics, Erasmus MC-Sophia Children's Hospital, Erasmus Medical Center Rotterdam, Rotterdam, The Netherlands, 3 Department of Sustainable Productivity and Employability, Netherlands Organization for Applied Scientific Research, Leiden, The Netherlands, 4 Research Institute of Child Development and Education, University of Amsterdam, Amsterdam, The Netherlands, 5 Department of Pediatrics, Willem-Alexander Children's Hospital, Leiden University Medical Center, Leiden, The Netherlands

¶ Membership of the Improve Research Network is provided in the Acknowledgments.
* l.solms@uva.nl

**Data Availability Statement:** The datafile supporting the paper will be available on OSF and

## Abstract

### Background

Physicians increasingly show symptoms of burnout due to the high job demands they face, posing a risk for the quality and safety of care. Job and personal resources as well as support interventions may function as protective factors when demands are high, specifically in times of crisis such as the COVID-19 pandemic. Based on the Job Demands-Resources theory, this longitudinal study investigated how monthly fluctuations in job demands and job and personal resources relate to exhaustion and work engagement and how support interventions are associated with these outcomes over time.

### Methods

A longitudinal survey consisting of eight monthly measures in the period 2020–2021, completed by medical specialists and residents in the Netherlands. We used validated questionnaires to assess job demands (i.e., workload), job resources (e.g., job control), personal resources (e.g., psychological capital), emotional exhaustion, and work engagement. Additionally, we measured the use of specific support interventions (e.g., professional support). Multilevel modeling and longitudinal growth curve modeling were used to analyze the data.

### Results

378 medical specialists and residents were included in the analysis (response rate: 79.08%). Workload was associated with exhaustion ($\gamma$ = .383, $p$ < .001). All job resources, as well as the personal resources psychological capital and self-judgement were associated with work engagement ($\gamma$s ranging from -.093 to .345, all $p$s < .05). Job control and

can be previewed here: https://osf.io/stfdp/?view_only=7bc333ca8cd04978936d2946a1760007.

**Funding:** The author(s) received no specific funding for this work.

**Competing interests:** The authors have declared that no competing interests exist.

psychological capital attenuated the workload-exhaustion relationship while positive feedback and peer support strengthened it (all $p$s < .05). The use of professional support interventions (from a mental health expert or coach) was related to higher work engagement (*estimate* = .168, $p$ = .032) over time. Participation in organized supportive group meetings was associated with higher exhaustion over time (*estimate* = .274, $p$ = .006).

## Conclusions

Job and personal resources can safeguard work engagement and mitigate the risk of emotional exhaustion. Professional support programs are associated with higher work engagement over time, whereas organized group support meetings are associated with higher exhaustion. Our results stress the importance of professional individual-level interventions to counteract a loss of work engagement in times of crisis.

## Introduction

Physicians around the globe are faced with high workload, time pressure, emotional demands, and an increasing clerical burden [1–3]. Moreover, they are ingrained in a system that tends to value perfectionism and lack of vulnerability over self-care and personal health. [1, 4, 5]. Not unexpectedly, physicians report increasing distress and symptoms of burnout [6, 7], such as mental and physical exhaustion and professional inefficacy [8]. Burnout is not only a heavy burden for individual physicians but can also have detrimental effects on the quality of medical care, including increased medical errors, [6, 9–12], and it may cause attrition among medical staff. Altogether, the costs of physician burnout are high for individuals, patients, and the healthcare system as a whole. To make matters worse, the COVID-19 pandemic has recently added to physicians' work demands, posing an amplified threat to their (mental) health [13]. Around the world, healthcare workers–particularly those on the frontline–have been greatly impacted both physically and mentally by the COVID-19 pandemic: staff shortages leading to long working hours, a lack of protective equipment, seeing many patients die, the risk of contracting COVID-19, and worries about colleagues, family, and friends have taken a toll on healthcare workers' mental health. Indeed, several studies found that healthcare workers' mental health has worsened during the pandemic, with many reporting symptoms of depression, anxiety, and distress [14–16], and an increasing risk of physician burnout [17]. In order to combat physician burnout and thus ensure patient safety [18] in times of crisis and beyond, an important first step is to understand which factors can protect physicians' well-being.

While high job demands have long been recognized as the prime factor leading to (physician) burnout [8, 19], job and personal resources have the opposite effect: they help respond to job demands and stimulate learning and development [20]. According to the Job Demands-Resources model (JD-R model) [8], resources are not just important in their own right but also buffer the negative effects of high job demands on burnout [21, 22]. Moreover, resources promote work engagement, a positive state characterized by dedication, vigor, and absorption, which bolsters performance and reduces the involvement in medical errors [23, 24]. In line with the JD-R model, research has shown that resources such as self-compassion, self-efficacy and development opportunities are associated with lower levels of exhaustion and higher levels of work engagement in medical professionals, including residents [25–28]. Yet, these findings are based on inter-individual (or between-person) designs that are prone to bias due to

individual confounds and ignore the fact that demands and resources can fluctuate within individuals over time. For example, physicians' workload may be higher during flu season, and their resources may change depending on a changing work context (e.g., change in experienced autonomy or support).

## An intra-individual perspective on physicians' exhaustion and work engagement

Especially in context of the COVID-19 pandemic, demands depend heavily on external factors such as fluctuating infection rates that are inevitably associated with hospital admissions. These oscillating work situations call for an intra-individual (or within-person) perspective in which job demands, resources, and outcomes fluctuate from month to month within the same person (see also Xanthopoulou et al., 2012) [29]. Hence, in this eight-wave longitudinal survey study, we adopt such an intra-individual perspective and investigate whether monthly job demands, job resources, and personal resources are associated with monthly exhaustion and work engagement (Fig 1). Specifically, we expect that monthly workload is positively associated with monthly exhaustion (Hypothesis 1), and that monthly job resources and personal resources are positively associated with monthly work engagement (Hypothesis 2a and 2b). Finally, we expect that monthly job and personal resources moderate the relationship between monthly workload and exhaustion in such a way that the relationship is weaker in months where physicians experience more resources (Hypothesis 3a and 3b).

## Interventions to support physician well-being

Understanding the effectiveness of support interventions is crucial to provide a timely remedy to physicians' distress. In times of high work demands, such as during the COVID-19 pandemic, physicians may need extra resources to cope with these demands. To support overburdened physicians, individual-level interventions may help [30–32]. Especially promising in this regard are professional support interventions, such as individual coaching, which has been shown to reduce burnout symptoms and strengthen the personal resources of healthcare professionals [33–35]. Other psychological support interventions, such as organized individual peer support (e.g., 'buddy' systems), organized supportive group meetings, and interventions aimed at improving collaborations and cohesion within teams, have been linked to physician well-being, also in the context of disease outbreaks [36, 37]. As such, these types of interventions may serve as effective strategies to protect the mental health of physicians during the COVID-10 pandemic.

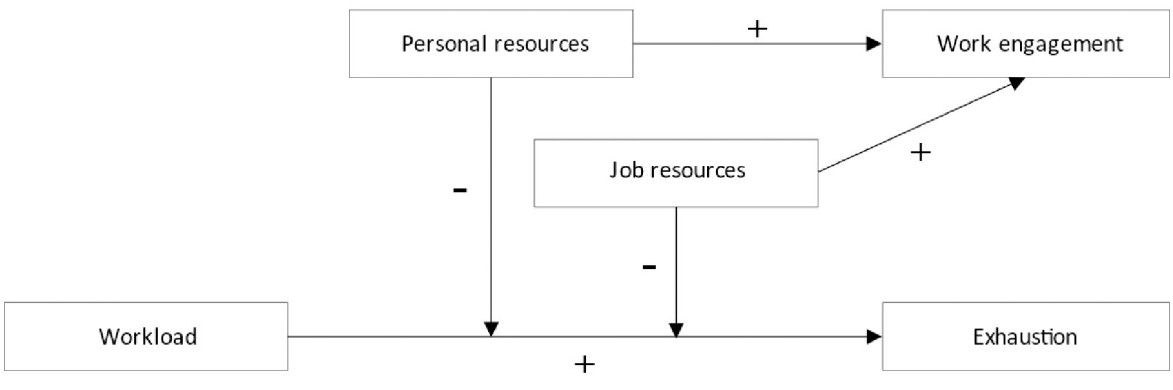

**Fig 1. Research model based on the JD-R model.**

Although these supportive and developmental interventions are recommended (for instance by the World Health Organization, see also Sheather & Slattery) [18], and implemented in healthcare organizations, it is unclear to what extent physicians make use of and value these interventions in demanding times. Perhaps more importantly, it remains unclear how these interventions relate to burnout and work engagement over time. Here we investigate if different interventions aimed at support and personal development (i.e., a class or workshop, an app or online information, individual peer support, group support, and professional support from a mental health expert or coach) can help to mitigate burnout and revive engagement [38–40]. We employ an intra-individual monthly diary design to explore whether these support interventions for physicians are associated with the development of exhaustion and engagement over time.

## Method

### Design and procedure

Data was collected from medical specialists and residents (including graduate physicians that were not in residential training) with various specializations and from diverse (academic and general) hospital and healthcare organizations throughout the Netherlands. Originally planned for 12 monthly measurements, data collection was finalized after eight measurement occasions (i.e., from June 2020 until March 2021) due to approaching data saturation and a decreasing number of respondents. The institutional Ethic Review Board of the University of Amsterdam approved this study on June 23, 2020; document 2020-WOP-12342. Written ethical approval was obtained from participants at the start of the study. See the description of the study sample for details.

### Measures

In line with the JD-R model we measured job demands (i.e., workload), job resources (i.e., managerial and peer support, job control, and positive feedback) and personal resources (i.e., self-judgement and psychological capital) as independent variables, and emotional exhaustion and work engagement as dependent variables. Additionally, we asked participants to report about specific interventions they had experienced in the preceding month. Finally, we assessed demographics and potential control variables.

**Independent and dependent variables.**   The (in)dependent variables were assessed at each wave with validated but shortened scales (T1-T8) and were scored on a 7-point scale ranging from 1 ('totally disagree') to 7 ('totally agree'). Because our study included multiple monthly waves during a period where healthcare workers were extremely strained, shortened scales were used to keep participants' taxation to a minimum and limit sample attrition. Initially, we asked participants to indicate if they had been absent due to vacation or sickness leave during the past seven working days. If participants had been absent, items measuring the (in)dependent variables referred to the past four weeks (i.e., 4-week survey version) instead of the past seven days (i.e., 7-days survey version; participants who had not recently been on leave, reported higher engagement). We included survey version as control variable in our analyses. This in order to prevent a distortion of people's answers due to the fluctuations in job demands and job and personal resources that are likely caused by holiday and sickness leave.

*Workload* was measured with 3 items from the Quantitative Workload Inventory [41] and one additional item that the researchers added to measure work-life conflict. An example item from the Quantitative Workload Inventory is: 'I experience emotional strain from my job' The additional item is: 'My personal life was under pressure from my work.'

*Managerial support* and *peer support* were measured with the same 2 items from the multi-dimensional scale of perceived social support [42] and one additional item added by the researchers. For peer support, the items referred to a colleague instead of a supervisor. An example item for managerial support is: 'I have experienced support from my supervisor.' An example item for peer support is: 'My colleagues tried to really help me.'

*Job control* was measured with 2 items from Jackson et al. [43] (see Solms et al., 2019) [25] and one additional item from the Work Design Questionnaire [44]. An example item is: 'I can set my own pace of work.'

*Positive feedback* was measured with 3 items, including 2 adapted items from the Work Design Questionnaire [44]. An example item is: 'I receive appreciation for my work from others.'

*Self-judgement* was measured with 4 items from the Self-compassion scale [45]. An example item is: 'When times are really difficult, I tend to be tough on myself.'

*Psychological capital* was measured with 4 items reflecting the four subscales of the construct, that is hope, optimism, resilience, and self-efficacy [46]. We used items that were previously used in a Dutch sample [47]. Because the original self-efficacy items refer to a managerial context [46], we used the developed items by Vink and colleagues [47] to measure general work-related self-efficacy that would fit the context of this study. All other items were part of the Psychological Capital Questionnaire [46]. An example item is: 'When encountering difficult problems at my work, I knew how to solve them.'

*Emotional exhaustion* was measured with 3 items from the Dutch version of the Maslach Burnout Inventory-General Survey [48, 49]. An example item is: 'I feel mentally drained from my work.'

*Work engagement* was measured with 3 items from the Utrecht Work Engagement Scale, with one item measuring each dimension (i.e., vigor, dedication, and absorption) [50]. An example item for the dimension of dedication is: 'I feel enthusiast about my work.'

**Assessment of intervention use.** Use of interventions was measured at each wave by asking participants if they had participated in the past four weeks in any of the following intervention programs offered to provide support and/or personal development: a class or workshop, an app or online information on the intranet, organized individual support from a peer, organized supportive group meetings, professional support from a mental health expert or coach (individual or in a group), support from an occupational physician, or any other kind of support (Table 1). The latter two options were included to be exhaustive but were excluded from further analyses because of limited use. Then, participants rated the usefulness of these interventions on a scale from 1 (very useless) to 10 (very useful), and indicated the topics discussed during the intervention.

**Table 1. Type of support interventions.**

| *Type of intervention* | *Example of content* |
| --- | --- |
| 1. A class or workshop | Management course, clinical knowledge |
| 2. An app or online information on the intranet | Mindfulness, COVID-19 updates |
| 3. Organized individual support from a peer | 'Buddy' meetings with assigned colleague |
| 4. Organized supportive group meeting | Intervision, debriefing |
| 5. Professional support from a mental health expert or coach[a] | Professional coaching, counseling |
| 6. Support from an occupational physician | - |

[a] Refers to individual or group-based programs.

**Demographics and control variables.** At T1, we measured gender, age, job position (specialist, resident), and fulltime employment as well as learning goal orientation and trait anxiety. Learning goal orientation refers to a person's 'preference to develop one's competence by acquiring new skills and mastering new situations' [51] and was measured because of its relevance for self-regulatory behaviors and its potential association with engagement and intervention use [52]. We measured learning goal orientation with 4 items from Vandewalle [51]. Trait anxiety refers to a stable proneness to experience state anxiety frequently [53, 54] and was measured because of its potential association with workload and exhaustion [55]. We used 4 items from the State-Trait Anxiety Inventory [56], and 2 additional items from the State-Trait Inventory for Cognitive and Somatic Anxiety [53]. At each wave, we measured survey version (i.e., 7-day version, 4-week version), contact with COVID-19 patients in the prior month (yes, no), and anxiety of COVID-19 infection (1 = not at all, 5 = very much), because of their potential associations with workload and exhaustion [55].

## Analytical approach

Because of the hierarchical structure of our data, we conducted hierarchical linear modeling (i.e., multilevel models) using M*plus* 7.31 [57]. For testing the hypothesized JD-R relationships, we performed a series of multilevel path analyses with emotional exhaustion and work engagement as dependent variables, in which the eight measurement times (level 1) were nested within individuals (level 2). See S1 Appendix for details about the path analyses.

To explore the use, usefulness, and the association of interventions with key study outcomes, we first examined the within- and between-person level correlations between the key study variables and the use of interventions. Second, we examined the extent to which participants found the interventions useful. Finally, we examined the associations between interventions and the trajectory of exhaustion and engagement over time with parallel process growth modeling using M*plus* statistical software [57]. Incomplete data points were excluded from the analyses. We examined several models (see results section) and used conventional model fit indices and growth components to estimate the fit of the models to the data.

## Results

### Study sample

Of all eligible participants who started the first survey, 478 participants completed it and gave consent. Due to an error in Qualtrics, three participants did not consent to but completed the survey; two of them were excluded in the analyses. Participants who did not complete the first survey were not invited for subsequent surveys. Because we were interested in the development of participants over time, we only included participants who had completed at least three of the eight surveys in the analyses (Fig 2). The final sample consisted of 378 participants (response rate: 79.08% of baseline sample; 74.1% women; 48.7% medical specialists, 51.3% medical residents), of which 50.3% completed all eight surveys. Mean age of participants was 38.6 years (10.41) and 41.3% of them (*n* = 156) indicated to work fulltime (Table 2).

### Preliminary analyses

**Multilevel structure.** We first assessed whether multilevel modeling was justified for our data by examining the within- and between-individual variance (i.e., the intraclass-correlation coefficient; ICC) for the model variables by computing intercept-only baseline models. Results showed that 47% of the variance in exhaustion, and 44% of the variance in engagement were

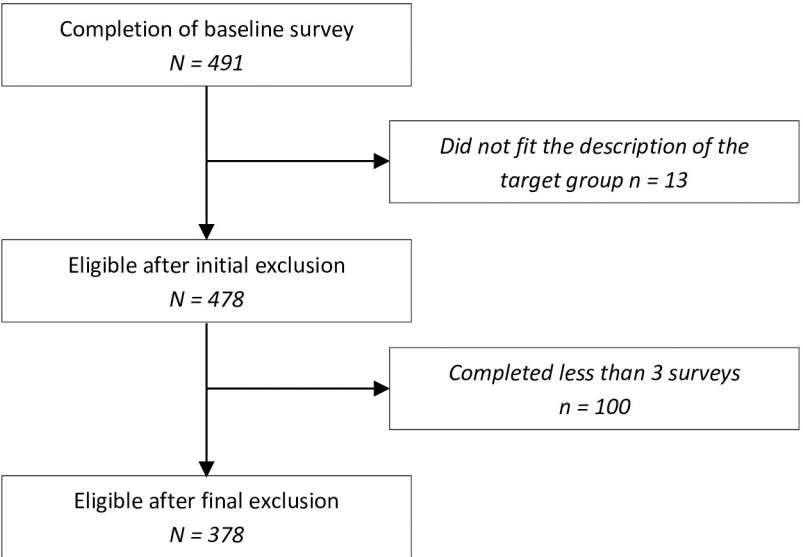

**Fig 2. Flowchart depicting exclusion procedure.**

attributable to within-person variations, justifying our multilevel approach. The ICCs for all study variables, displayed in S1 Table, varied from 46.4% to 59.2%.

**Control variables.** Prior to testing the hypotheses, we estimated the associations of our potential control variables with both dependent variables (see Table 3, control model). We only included those variables that showed significant correlations with either exhaustion or work engagement (see S1 Table). Based on these results (described in S2 Appendix), we

**Table 2. Demographics of residents and specialists participating in a longitudinal survey study June 2020 until March 2021.**

| Characteristics | Residents | Specialists |
|---|---|---|
| | No (% of 194) | No (% of 184) |
| Gender | | |
| Female | 155 (79.9%) | 125 (67.9%) |
| Male | 39 (20.1%) | 59 (32.1%) |
| Age[a,b] | | |
| 20–30 years | 84 (43.3%) | 1 (0.5%) |
| 31–40 years | 107 (55.2%) | 60 (32.6%) |
| 41–50 years | 2 (1.0%) | 59 (32.1%) |
| 51–60 years | | 47 (25.5%) |
| 61 years and older | | 17 (9.2%) |
| Fulltime employment[c] | 93 (47.9%) | 63 (34.2%) |
| Home situation | | |
| Cohabitation[c,d] | 161 (83.0%) | 158 (85.9%) |
| Children[c], one or more | 34 (17.5%) | 116 (63.0%) |

[a] Due to rounding, the overall percentage can slightly deviate from 100%.

[b] Percentage values include missing values (*n* = 1) for residents.

[c] Percentage values include missing values (*n* = 1) for specialists.

[d] with partner or other.

**Table 3. Multi-level multiple regression of emotional exhaustion and work engagement on workload, job resources, personal resources and the workload x resources interaction terms.**

| Predictors | Emotional exhaustion (B) | | | | Work engagement (B) | | | |
|---|---|---|---|---|---|---|---|---|
| | Null model | Control model | Main effect model | Interaction model | Null model | Control model | Main effect model | Interaction model |
| Intercept | 2.95*** | 2.99*** | 2.87*** | 2.86*** | 5.29*** | 5.09*** | 5.07*** | 5.07*** |
| *Level 2 variables* | | | | | | | | |
| Gender[a] | - - | -0.086 | - - | - - | - - | -0.003 | - - | - - |
| Job position[b] | - - | 0.014 | - - | - - | - - | 0.080 | - - | - - |
| Learning goal Orientation | - - | -0.045 | -0.040 | -0.038 | - - | 0.314*** | 0.308*** | 0.308*** |
| Trait anxiety | - - | 0.474*** | 0.472*** | 0.472*** | - - | -0.269*** | -0.278*** | -0.277*** |
| *Level 1 variables* | | | | | | | | |
| Time | - - | 0.050* | 0.030 | 0.033 | | -0.020 | -0.005 | -0.005 |
| Survey version[c] | - - | -0.027 | -0.007 | -0.005 | - - | 0.098*** | 0.078*** | 0.078*** |
| Infection anxiety | | 0.068** | 0.023 | 0.022 | | -0.040* | -0.025 | -0.025 |
| Workload | - - | - - | 0.383*** | 0.380*** | - - | - - | -0.046* | -0.046* |
| Managerial Support | - - | - - | -0.012 | -0.007 | - - | - - | 0.054** | 0.054** |
| Peer support | - - | - - | -0.012 | -0.016 | - - | - - | 0.081*** | 0.081*** |
| Job control | - - | - - | -0.078*** | -0.078*** | - - | - - | 0.041* | 0.041* |
| Positive Feedback | - - | - - | -0.099*** | -0.100*** | - - | - - | 0.345*** | 0.345*** |
| Self-judgment | - - | - - | 0.096*** | 0.098*** | - - | - - | -0.093*** | -0.093*** |
| Psych. capital | - - | - - | -0.088*** | -0.085*** | - - | - - | 0.187*** | 0.187*** |
| Workload*man.support | - - | - - | - - | -0.017 | - - | - - | - - | - - |
| Workload*peer support | - - | - - | - - | 0.049* | - - | - - | - - | - - |
| Workload*control | - - | - - | - - | -0.061** | - - | - - | - - | - - |
| Workload*feedback | - - | - - | - - | 0.068** | - - | - - | - - | - - |
| Workload*judgement | - - | - - | - - | -0.008 | - - | - - | - - | - - |
| Workload*psycap | - - | - - | - - | -0.090*** | - - | - - | - - | - - |
| -2 log likelihood | 14080.72 | 13591.064 | 11981.73 | 11944.456 | 14080.72 | 13591.064 | 11981.73 | 11944.456 |

[a] 0 = female, 1 = male;

[b] 0 = resident, 1 = medical specialist;

[c] 0 = 1-month version, 1 = 7-day version.

Measurement time runs from 0 to 7. The level 2 variables are grand-mean centered. The level 1 variables are person-mean centered.

*$p < .05$,

**$p < .01$.

***$p < .001$ (all 2-tailed).

We report the standardized regression coefficients and intercepts.

included time, survey version, anxiety of COVID-19 infection, trait anxiety as well as learning goal orientation–all of which showed relationships with exhaustion and/or work engagement–as control variables in the model testing. The control model showed significant improvement in model fit over the null model ($\Delta\chi^2 = 489.656$, $\Delta df = 14$, $p < .001$).

## Hypotheses testing

**Direct effects.** H1 stated that workload would be positively associated with exhaustion. The main effects model (Table 3) supported this hypothesis: workload was positively related to exhaustion ($\gamma = .383$, $p < .001$). H2 stated that job resources (H2a) and personal resources (H2b) would be positively associated with engagement. Results supported H2a in that all four

job resources were positively related to engagement ($\gamma$s ranging from .041 to .345, all $ps < .05$), and H2b in that psychological capital was positively related to engagement ($\gamma = .187$, $p < .001$), and self-judgement was negatively related to engagement ($\gamma = -.093$, $p < .001$). The main effect model resulted in a significant improvement in explained variance of our outcomes over the control model ($\Delta\chi^2 = 1609.334$, $\Delta$df = 10, $p < .001$).

**Interaction effects.** H3a stated that job resources would moderate the relationship between workload and exhaustion. The interaction model (Table 3) revealed that the interaction terms for job control ($\gamma = -.061$, $p = .003$), positive feedback ($\gamma = .068$, $p = .003$), and peer support ($\gamma = .049$, $p = .018$) but not for managerial support were significantly related to exhaustion. Simple slope tests (with high and low values referring to values 1 SD above and below the mean, respectively; here we report the unstandardized estimates for the new parameters) revealed that the positive link between workload and exhaustion was weaker when job control was high ($\gamma = .392$, $p < .001$) rather than low ($\gamma = .512$, $p < .001$); stronger when feedback was high ($\gamma = .522$, $p < .001$) rather than low ($\gamma = .382$, $p < .001$); and stronger when peer support was high ($\gamma = .503$, $p < .001$) rather than low ($\gamma = .401$, $p < .001$). These results indicate that the negative impact of workload on exhaustion was buffered by more job control and amplified by more positive feedback and peer support. Therefore, H3a was not supported. H3b stated that personal resources would moderate the relationship between workload and exhaustion. The interaction model (Table 3) revealed that the interaction term for psychological capital was negatively related to exhaustion ($\gamma = -.090$, $p < .001$), while the interaction term for self-judgement was unrelated to exhaustion. A simple slope test revealed that the positive link between workload and exhaustion was weaker when psychological capital was high ($\gamma = .368$, $p < .001$) rather than low ($\gamma = .536$, $p < .001$) (Fig 3), indicating that more psychological capital, but not lower self-judgment, can buffer the negative impact of workload on exhaustion. Thus, H3b was partly supported. The interaction model resulted in a small but significant improvement in explained variance over the main effects model ($\Delta\chi^2 = 32.27$, $\Delta$df = 6, $p < .001$).

## Interventions: Use, usefulness, and associations with key outcomes

One of the key goals of this study was to understand how participation in support programs relates to physicians' well-being over time. To that end, we assessed participants' use of interventions, the extent to which they perceived interventions as useful, and how interventions related to exhaustion and work engagement.

**Use of interventions.** Across the eight measurements, participants most frequently participated in a course/workshop, and organized supportive group meeting (see S2 Table). From T1 to T4, participants also frequently used online information/app, while from T5 to T8, participants more frequently sought professional support as compared to using information/app or organized individual peer support. Next, we explored whether participation in specific interventions was associated with workload, job and personal resources, exhaustion, and work engagement. Both the between-level and within-level results (see S3 Table) indicate that participants with relatively higher demands (i.e., workload), lower resources (e.g., lower job control, more self-judgement), and more symptoms of exhaustion were more likely to use support interventions such as organized individual peer support, organized supportive group meetings, and professional support. See S3 Appendix for a detailed description of those associations.

**Usefulness of interventions.** Next, we explored the extent to which participants experienced specific interventions as useful. To this end, we aggregated the usefulness scores across measurements. As displayed in S2 Table, participants perceived professional support ($M = 8.00$, $SD = 1.31$) and organized individual peer support ($M = 7.92$, $SD = 1.25$) as most

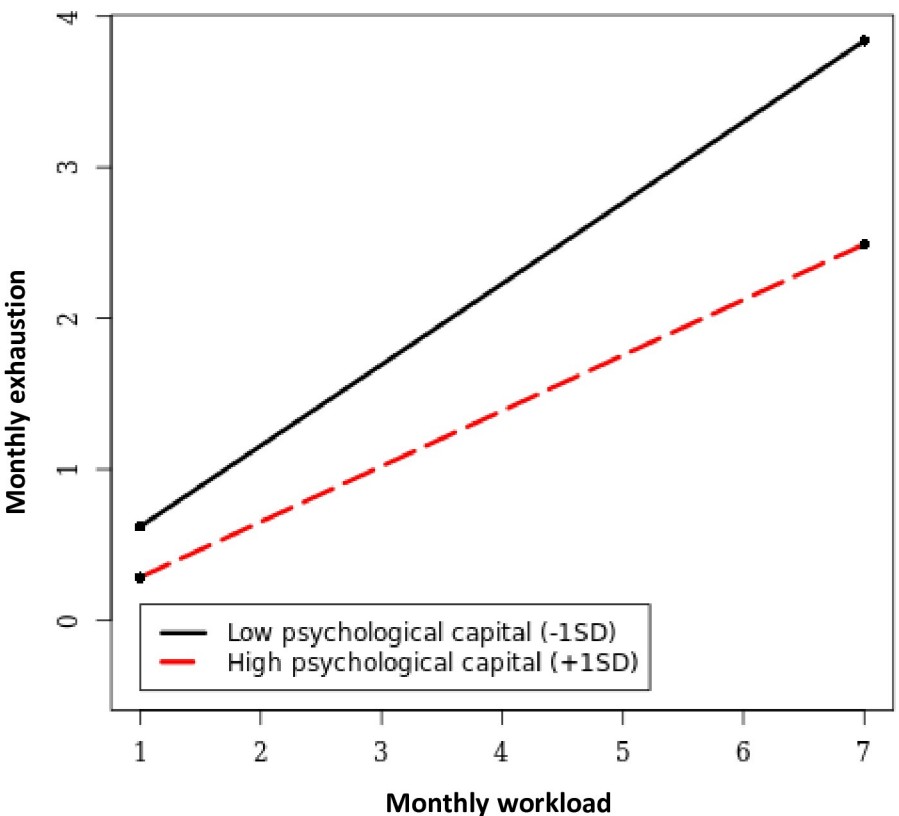

**Fig 3. Psychological capital as moderator of the workload-exhaustion relationship.**

useful in comparison to an organized supportive group meeting ($M$ = 7.46, $SD$ = 1.14), a course/workshop ($M$ = 7.53, $SD$ = 1.04) and online information/app ($M$ = 6.78, $SD$ = 1.51).

**Intervention use: Associations with exhaustion and work engagement.** To explore the association between intervention use and exhaustion and work engagement, we tested several models. See S4 Appendix for information on model testing, the growth factor estimates (i.e., intercept and slopes), and the model fit of the initial parallel process growth model excluding any predictors. Except for the slope factor means and their variance, we report the standardized estimates. In model 1, we examined the associations between all potential control variables and the intercepts (i.e., starting values at T1) and slopes (i.e., trajectories reflecting change over time) of exhaustion and work engagement. 376 participants were included in the model due to missing values on the control variables. The model fit was good, CFI = .953, TLI = .945, RMSEA = .043. Analyses revealed that: (1) trait anxiety and anxiety of COVID-19 infection were associated with higher mean levels of exhaustion and lower mean levels of work engagement at T1, (2) learning goal orientation and job position were associated with higher mean levels of work engagement at T1, (3) trait anxiety was associated with change in (i.e., the slope of) exhaustion, and (4) age, learning goal orientation, and anxiety of COVID-19 infection were associated with change in work engagement. See S4 and S5 Tables for details on these estimates. These variables were therefore included in the second model, in which we explored the associations between professional support and the key study variables. For each intervention category, we calculated a mean score consisting of the number of times that a participant took part in the intervention, divided by the number of completed surveys (see note, S3 Table). Model 2 and 3 included 377 participants due to a missing value on one of the control variables.

In model 2, we estimated how relatively higher (vs. lower) mean levels (i.e., intercepts) of exhaustion and work engagement were associated with the use of professional support and how the use of professional support, in turn, was related to change in (i.e., the slopes of) exhaustion and work engagement. The model fit the data well, CFI = .964, TLI = .958, RMSEA = .041. Estimates for the covariances between growth factors (see S4 Appendix) were largely identical to the initial model: Results showed that the intercepts of exhaustion and work engagement ($r$ = -.554, $p$ < .001), and the slopes of both constructs ($r$ = -.60, $p$ < .001) were negatively associated: higher levels in exhaustion were associated with lower levels in work engagement (and vice versa), and greater increases in exhaustion over time were associated with smaller increases in engagement (and vice versa). The covariance between the intercept and slope of work engagement–but not of exhaustion–was negative ($r$ = –.235, $p$ = .016) indicating that higher initial levels in work engagement were associated with smaller slope values. Furthermore, higher initial levels of exhaustion and work engagement (i.e., intercepts) did not predict the use of professional support interventions (*estimate* = .104, $p$ > .05, *estimate* = -.031, $p$ > .05, for exhaustion and work engagement, respectively). The use of professional support interventions, in turn, was not associated with the slope of exhaustion (*estimate* = -.131, $p$ = .117) but with the slope of engagement (*estimate* = .171, $p$ = .029). These results indicate that an increase in professional support was associated with an increase in the slope of engagement, signaling that people who participated in professional support interventions reported a relative improvement in their level of work engagement over time.

In model 3, we explored the associations between alternative interventions (i.e., course/ workshop, online information/app, organized individual peer support, organized supportive group meeting) and the key study variables. To this purpose, we examined the association between all five types of interventions and the intercepts and slopes of exhaustion and work engagement simultaneously. We included covariances between interventions in the model that reached significance. The model fit the data well, CFI = .966, TLI = .956, RMSEA = .036. Results for intercept-intercept, slope-slope as well as intercept-slope covariation were largely identical to the results presented in model 2 (see S4 and S5 Tables). The slope factor mean for exhaustion was, contrary to the initial models, significant in this final model (*estimate* = 0.022, $p$ = .037), indicating an average increase in exhaustion over time. The results for professional support were comparable to those reported in model 2 but extended insofar that higher levels of exhaustion and work engagement predicted more participation in an organized individual peer support intervention (*estimate* = .297, $p$ = .004; *estimate* = .250, $p$ = .006 for exhaustion and work engagement, respectively). Participation in an organized supportive group meeting was associated with the slope of exhaustion (*estimate* = .274, $p$ = .006), indicating that an increase in participation in organized supportive group meetings was associated with an increase in the slope of exhaustion, signaling that people who participated in supportive group meetings reported a relative deterioration of exhaustion over time.

## Discussion

To prevent physician burnout, it is crucial to uncover the factors that thwart and support physician well-being in daily practice [30, 58], and to understand how interventions can contribute to these factors in times of increased stress, such as during the COVID-19 pandemic [30, 31, 59–62]. In this study, we took an intra-individual (i.e., within-level) perspective and surveyed a sample of physicians for eight consecutive months to deepen our understanding of how fluctuations in job demands and job- and personal resources related to their feelings of exhaustion and work engagement. In addition, we explored the associations between multiple support programs and physicians' exhaustion and work engagement.

Our study revealed that, consistent with JD-R theory and previous findings [63–65], exhaustion was higher in months when workload was higher, and work engagement was higher in months when job and personal resources were higher. Furthermore, in support of the 'buffer'-hypothesis [21], the resources control and psychological capital attenuated the negative effects of workload on exhaustion. Positive feedback and peer support however strengthened this relationship. Furthermore, physicians with relatively higher demands, lower resources, and more symptoms of exhaustion were more likely to use person-oriented support interventions (e.g., organized individual peer or professional support). Additionally, we found that professional support interventions were associated with a relative improvement in work engagement, while organized group support interventions were associated with a relative deterioration of exhaustion.

## Resources that buffer or intensify exhaustion

Our finding that positive feedback and peer support strengthened the workload-exhaustion relationship is surprising given that common stress models unanimously predict positive effects of resources on stress and well-being [8, 66, 67]. We see three possible explanations for this finding. First, it is possible that peer support, where physicians share their concerns with each other, induces rather than reduces their emotional state. This phenomenon, in which the (negative) emotions of the people around us rub off and cause similar emotional experiences, is called emotional contagion [68] and might explain why peer support amplified emotional exhaustion when physicians experienced high workload. This phenomenon might also explain our finding that organized group support interventions lead to increases in exhaustion.

Second, job resources may only (or particularly) be helpful when they are functional for the task at hand. This idea is based on the Demand-Induced Strain Compensation Model [69] that states that the adverse effects of high job demands can best be countered through functional, *corresponding* types of job resources. Based on this 'matching principle', physicians' high workload might best be countered with resources that reduce physical and cognitive load (i.e., help to pursue goals, increase self-efficacy or autonomy) [70] rather than 'social' job resources (e.g, feedback and peer support) [71].

Third, it is possible that positive feedback can incentivize effort and hardiness because people tend to protect their resources and aim to uphold their positive self-views [67, 72]. Notably, although positive feedback strengthened rather than weakened the workload-exhaustion relationship, it also showed to have a direct and negative relationship with exhaustion and a positive one with work engagement. This suggests that physicians who experience positive feedback are generally less prone to emotional exhaustion and more engaged. At the same time, the beneficial role of positive feedback for emotional exhaustion diminishes as workload increases.

## A problem shared is (not always) a problem halved

Physicians were more inclined to use person-oriented support interventions such as individual peer support, group peer support, and professional support when they had fewer resources. They also experienced individual peer support and professional support interventions as most useful. This finding is in line with previous intervention studies indicating that interventions are taken up especially in times when resources are low and strain is high [35, 73]. Yet, only professional support interventions contributed to physicians' well-being by increasing work engagement, while organized group support interventions further increased exhaustion. The latter finding is unexpected, but in line with our finding that peer support worsened physicians' well-being when experiencing high workload. That is, although venting to peers might

feel liberating and relieving in the short run, it might not be functional because this emotional support does not necessarily lead to actual changes and solutions that improve physicians' well-being and can even induce emotional contagion: the negative emotions of peers may spill over, may bring down individual team members, and may trigger their further depletion [68, 74]. Thus, professional support interventions are preferable over peer support interventions, as professional interventions provide individuals with both emotional support and task support by helping them to develop (cognitive or behavioral) strategies to solve their problems in a tangible and sustainable way [75]. This is in line with previous findings indicating that receiving professional support (including goal-setting and action-planning) helped physicians to strengthen their personal resources and increase work engagement [34].

## Strengths and limitations

A major strength of this study is its multi-wave within-level design, allowing for a realistic display of the fluctuations in physicians' work environment amid the COVID-19 pandemic. Moreover, this design reduces Type II errors and confounds associated with individual differences. As such, this strong methodological approach extends existing knowledge and can form an excellent basis for designing much called-for effective interventions to prevent and reduce physician burnout and promote work engagement [6], both of which have been linked to quality of care [23].

However, our study also has limitations. Although our data showed support for the hypothesized relationships, reversed or reciprocal relationships remain possible. For example, job resources may influence work engagement while work engagement may in turn also foster job resources [76, 77]. Such reciprocal relationships may be tested with experience sampling designs examining cross-lagged relationships, but it should be noted that definite causal conclusions can only be drawn from experimental designs [78, 79]. Another limitation is the predominantly female sample in this study, although the high percentage of female physicians is representative of the Dutch labor market. Even though we controlled for gender in our analyses, our findings may be less generalizable to male physicians. We also note that the focus of this study was on emotional exhaustion, rather than the complete burnout syndrome. Future studies could extend these findings by including the two remaining facets of burnout, depersonalization and personal accomplishment. Furthermore, we used shortened scales for all study variables due to practical constraints. We recognize that it would have been preferred to use the original scales, however, we expect that using shortened scales–that all showed good reliability–did not impact our results (see also Matthew et al., 2022; Fisher et al., 2016) [80, 81]. A final limitation is that our study does not differentiate between physicians at different career stages. Because attending and junior physicians may resort to different resources when facing high job demands [25], it is important to examine and–when necessary–customize interventions to the needs of specific target groups.

## Practical implications

Understanding which personal and job factors protect or jeopardize physician well-being is a prerequisite for designing effective interventions. Based on our findings, we advise to implement interventions that predominantly foster autonomy and psychological capital, both resources that can mitigate the negative impact of high workload on emotional exhaustion. At the same time, our results stress the importance of both job and personal resources for protecting physicians' work engagement in times of crisis. Yet, we need to note that physicians who experience high job demands tend to seek the support from their peers, while such support from peers can pose additional demands and thus may worsen employee well-being.

The finding that professional support interventions were associated with higher work engagement but not lower exhaustion has several important implications. Medical centers often implement a myriad of individual-level interventions (e.g., mindfulness, stress-management; for a review see Panagioti et al., 2017) [31] aimed to boost physician resilience and vitality. In practice, the effects of these interventions are rarely assessed. While our findings suggest that professional support programs cannot halt or mitigate physicians' exhaustion in times of a structurally increased workload, such as during the COVID-19 pandemic, they do halt an erosion of work engagement. Our findings suggest that hospital organizations could invest more resources into individual support programs such as professional coaching, which has shown to boost physicians' personal resources (e.g., psychological capital) and improve their well-being, including work engagement [34, 35]. Additionally, the effects of these professional individual interventions may translate to the team level as it may help to cultivate an open and psychologically safe team culture, which benefits patient care [82, 83].

## Conclusion

Employing a multi-wave within-level design, this study showed that professional support programs show promise in strengthening work engagement, even in periods of increased stress such as the COVID-19 pandemic. Organized group support meetings were associated with higher rather than lower exhaustion. Personal and job resources such as psychological capital and job control can help physicians manage the demands of their work. Although system-level changes are needed to tackle physician burnout, professional person-centered interventions rather than social exchanges with peers can provide timely solutions to sustain physicians' work engagement in times of crisis.

## Supporting information

**S1 Appendix. Path analyses.**
(DOCX)

**S2 Appendix. Associations between potential control variables and exhaustion and work engagement.**
(DOCX)

**S3 Appendix. Associations between intervention use and key study variables.**
(DOCX)

**S4 Appendix. Associations between interventions and exhaustion and work engagement—model testing and growth factor estimates.**
(DOCX)

**S1 Table. Means, standard deviations, intra-class correlations, and intercorrelations among all study variables.**
(DOCX)

**S2 Table. Frequencies of intervention involvement across the study period.**
(DOCX)

**S3 Table. Intercorrelations among key study variables and intervention use.**
(DOCX)

**S4 Table. Results of growth modeling for exhaustion.**
(DOCX)

**S5 Table. Results of growth modeling for work engagement.**
(DOCX)

## Acknowledgments

The Improve Research Network: Suzanne Booij, Hein Brackel, Bianca Buurman, Laila B. van der Heijden, Ian Leistikow, Jan M.M. van Lith, Steffi Rombouts, Ralph K.L. So, Hanneke Verheijde.

The authors wish to thank all the medical residents and attending physicians who participated in this study.

## Author Contributions

**Conceptualization:** Lara Solms, Annelies E. M. van Vianen, Jessie Koen, Kees-Jan Kan, Matthijs de Hoog, Anne P. J. de Pagter.

**Data curation:** Lara Solms, Annelies E. M. van Vianen, Jessie Koen, Matthijs de Hoog, Anne P. J. de Pagter.

**Formal analysis:** Lara Solms, Annelies E. M. van Vianen, Jessie Koen, Kees-Jan Kan.

**Investigation:** Lara Solms, Annelies E. M. van Vianen, Jessie Koen, Matthijs de Hoog, Anne P. J. de Pagter.

**Methodology:** Lara Solms, Annelies E. M. van Vianen, Jessie Koen, Kees-Jan Kan, Matthijs de Hoog, Anne P. J. de Pagter.

**Supervision:** Annelies E. M. van Vianen, Jessie Koen, Matthijs de Hoog, Anne P. J. de Pagter.

**Writing – original draft:** Lara Solms, Annelies E. M. van Vianen, Jessie Koen, Kees-Jan Kan, Matthijs de Hoog, Anne P. J. de Pagter.

**Writing – review & editing:** Lara Solms, Annelies E. M. van Vianen, Jessie Koen, Kees-Jan Kan, Matthijs de Hoog, Anne P. J. de Pagter.

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
