## [Decision Letter · Decision Letter 0]

16 May 2022

PONE-D-22-07749Physician Exhaustion and Work Engagement During the COVID-19 Pandemic: The Role of Resources and Support InterventionsPLOS ONE

Dear Dr. Solms,

Thank you for submitting your manuscript to PLOS ONE. After careful consideration, we feel that it has merit but does not fully meet PLOS ONE’s publication criteria as it currently stands. Therefore, we invite you to submit a revised version of the manuscript that addresses the points raised during the review process.

**ACADEMIC EDITOR: **Dear Authors

Thank you very much for your submission and the great effort put in the paper presented. 

We would be able to proceed further once reviewers and editor suggesions are adressed.**Academic Editor suggestions:**The design chosen does not allow for actual effectiveness estimations. However, if considering comparative effectiveness from repeated-measurements observational data, highly we encourage to consider adding propensity scores. Please avoid the use of the section interventions in the methods sections since not facing here a clinical trial or to be describe differently. Same goes for the effectiveness in the results section. Consider including the study design in the title and rephrase it. Pay attention to table 1 suggestion from reviewer. Please consider moving some of the vast statistics table to the supplementary files also. Consider de use of a flow diagram. Ensure to adapt STROBE to the CHERRIES (https://pubmed.ncbi.nlm.nih.gov/15471760/) to have the full view of the questionnaire process. 

Best regards

We look forward to receiving your revised manuscript.

Kind regards,

Juan A López-Rodríguez

Academic Editor

PLOS ONE

Journal Requirements:

Reviewers' comments:

Reviewer's Responses to Questions

**Comments to the Author**

1. Is the manuscript technically sound, and do the data support the conclusions?

Reviewer #1: Yes

Reviewer #2: Yes

2. Has the statistical analysis been performed appropriately and rigorously? 

Reviewer #1: Yes

Reviewer #2: Yes

3. Have the authors made all data underlying the findings in their manuscript fully available?

Reviewer #1: Yes

Reviewer #2: Yes

4. Is the manuscript presented in an intelligible fashion and written in standard English?

Reviewer #1: Yes

Reviewer #2: Yes

5. Review Comments to the Author

Reviewer #1: This is an original article investigating the role of resources and support Interventions in the development of work exhaustion among physicians during the COVID-19 pandemic. The study presents the results of original psychometric research in an intelligible fashion and is written in standard English. The methodology is explained in detail and is appropriate for this type of research. Statistical analyses are performed to a high technical standard and are described in sufficient detail. There are no ethical issues to be addressed. Conclusions are presented in an appropriate fashion and are supported by the data. My opinion is that this paper should be accepted for publication without further corrections.

Reviewer #2: General comments: The thematic of the research is interesting, and it was well conducted. However, adjustments in the writing of the article must be made before approval.

Introduction

• I felt the lack of a focus on the scenario of COVID-19 as an aggravating factor for the exhaustion of professionals, and make it clearer in the introduction which support interventions will be studied.

Methods

• I suggest removing the entire description of the population, including the table, since they are results, from the second paragraph of the methodology.

Results

• The description of the results is very extensive, and difficult to read. I suggest the authors to be more direct and clear in the results; a detailed description of the statistical issue isn’t necessary.

• Focus on how to support interventions would improve this situation, which is the objective of the study.

Discussion:

• I consider important to cite other studies carried out on the subject and to make comparisons with their results.

6. PLOS authors have the option to publish the peer review history of their article (what does this mean?). If published, this will include your full peer review and any attached files.

Reviewer #1: No

Reviewer #2: No

---

## [Author Response · Author response to Decision Letter 0]

30 Jun 2022

The response letter to both the editor and the reviewers is uploaded in the 'attach files' section.

---

## [Decision Letter · Decision Letter 1]

9 Sep 2022

PONE-D-22-07749R1Physician Exhaustion and Work Engagement During the COVID-19 Pandemic: A Longitudinal Survey into the Role of Resources and Support InterventionsPLOS ONE

Dear Dr. Solms,

Thank you for submitting your manuscript to PLOS ONE. After careful consideration, we feel that it has merit but does not fully meet PLOS ONE’s publication criteria as it currently stands. Therefore, we invite you to submit a revised version of the manuscript that addresses the points raised during the review process.

ACADEMIC EDITOR:Thank you very much for this second effort in trying to address all reviewers comments. We finnaly had a third reviewer also having a look to your work who thinks it is a good job although some minor comments from the reviewer need to be review. We highly appreciate that you'd try discuss the comments regarding our second reviewer and be back to us as soon as possible. Best regardsJALRAcademic Editor

We look forward to receiving your revised manuscript.

Kind regards,

Juan A López-Rodríguez

Academic Editor

PLOS ONE

Journal Requirements:

Reviewers' comments:

Reviewer's Responses to Questions

**Comments to the Author**

1. If the authors have adequately addressed your comments raised in a previous round of review and you feel that this manuscript is now acceptable for publication, you may indicate that here to bypass the “Comments to the Author” section, enter your conflict of interest statement in the “Confidential to Editor” section, and submit your "Accept" recommendation.

Reviewer #1: All comments have been addressed

Reviewer #3: (No Response)

2. Is the manuscript technically sound, and do the data support the conclusions?

Reviewer #1: Yes

Reviewer #3: Partly

3. Has the statistical analysis been performed appropriately and rigorously? 

Reviewer #1: Yes

Reviewer #3: Yes

4. Have the authors made all data underlying the findings in their manuscript fully available?

Reviewer #1: Yes

Reviewer #3: Yes

5. Is the manuscript presented in an intelligible fashion and written in standard English?

Reviewer #1: Yes

Reviewer #3: Yes

6. Review Comments to the Author

Reviewer #1: There are no issues to be addressed. My opinion is that this paper should be accepted for publication without further corrections.

Reviewer #3: Physician Exhaustion and Work Engagement During the COVID-19 Pandemic: A

Longitudinal Survey into the Role of Resources and Support Interventions

Based on the Job Demands-Resources theory, this longitudinal study investigated how monthly fluctuations in job demands and job and personal resources relate to exhaustion and work engagement and how support interventions are associated with these outcomes over time.

The hypotheses raised were that monthly workload was positively associated with monthly exhaustion (Hypothesis 1), and that monthly work resources and personal resources were positively associated with monthly work engagement (Hypothesis 2a and 2b). Finally, the authors hypothesized that monthly work and personal resources moderate the relationship between monthly workload and exhaustion in such a way that the relationship was weaker in months when physicians had more resources (Hypothesis 3a and 3b).

This is an interesting study, with important results and that adds to what already exists in the literature on this topic.

The authors included 378 medical specialists and residents in the analysis and obtained a good response rate (79.08%).

The authors said they used validated questionnaires to assess job demands (i.e., workload), job resources (e.g., job control), personal resources (e.g., psychological capital), emotional exhaustion, and work engagement. However, they chose to use only part of these questionnaires and, for example, the questionnaire that measures Burnout was not used. This is a major limitation of the study and I would like the authors to explain why this choice, if this is validated, because it put your results at risk.

Burnout is widely described in its tridimensionality: (1) emotional exhaustion, (2) depersonalization and (3) personal accomplishment. The emotional exhaustion subscale (nine items) assesses feelings of being emotionally overextended and exhausted by work. The MBI is a 22-item questionnaire that has shown to be reproducible and valid . The MBI evaluates three domains of burnout.

“Emotional exhaustion was measured with 3 items from the Dutch version of the Maslach Burnout Inventory-General Survey. An example item is: ‘I feel mentally drained from my work.’”

According to the authors, the emotional exhaustion burnout dimension was measured with 3 items from the Dutch version of the Maslach Burnout Inventory-General Survey. Why did the authors assess separately and not use, as usual, the complete questionnaire to assess Burnout? Personally, I've never seen it used separately. Can the authors explain whether this is validated? Better to put in the limitation of the study that Burnout was not measured as well as the others questionnaires.

“Work engagement was measured with 3 items from the Utrecht Work Engagement

Scale.”

Work engagement is considered the core factor that affects various outcomes at the organizational and individual levels including absenteeism, turnover rate, profitability, and productivity.

Work engagement was assessed by the 9-item Utrecht Work Engagement Scale (UWES-9). The UWES-9 was hypothesized to measure work engagement in three dimensions: vigor (3 items), dedication (3 items) and absorption (3 items). The items are scored on a 7-point Likert scale ranging from 0 (never) to 6 (every day).

“We used 4 items from the State-Trait Anxiety Inventory, and 2 additional items from the State-Trait Inventory for Cognitive and Somatic Anxiety”.

A common measure of state and trait anxiety, the State-Trait Anxiety Inventory is used extensively in psychological research. The STAI consists of two identical 20-item subscales: one measuring state anxiety and the other measuring trait anxiety. For the state scale, individuals are asked to rate their anxiety "in the moment" and for the trait scale, individuals are asked to rate their anxiety "in general".

According to your results, “workload was associated with exhaustion. All works resources, as well as personal resources, psychological capital and self-judgment were associated with work engagement”. This is an important result, a protective factor for Burnout and moral distress. I believe it can be better discussed, a proposal for better engagement and prevention of moral stress.

Job and personal resources can safeguard work engagement and mitigate the risk of emotional exhaustion. Also mitigate a risk of moral distress, according to literature. The literature shows a clear association between Burnout and moral distress. Although it is not the scope of this study, the authors could discuss its results (such as Work and personal resources) as prevention of moral distress as well. See the study by Fumis et al., 2017, for example.

Severe burnout syndrome is present in all critical care providers. A positive relationship was found between burnout and moral distress, and after regression analysis, moral distress was independently associated with burnout. (Fumis et al. Ann. Intensive Care (2017) 7:71).

“Our results reinforce the importance of professional interventions at the individual level in the fight against medical burnout”. Did the authors think of any suggestions? In view of the high prevalence of burnout among intensivists and, especially, after the pandemic, I believe that the authors can highlight some suggestions for greater engagement as a force to prevent burnout.

7. PLOS authors have the option to publish the peer review history of their article (what does this mean?). If published, this will include your full peer review and any attached files.

Reviewer #1: No

Reviewer #3: No

---

## [Author Response · Author response to Decision Letter 1]

17 Oct 2022

Dear dr. Juan A López-Rodríguez,

Please find attached our revised paper entitled: “Physician exhaustion and work engagement during the COVID-19 pandemic: A longitudinal survey into the role of resources and support interventions” for possible publication in PloS One. 

We would like to thank you and the reviewer for the valuable input on our manuscript. We have revised the manuscript based on the suggestions of the third reviewer. The response to each of the comments, including how and where the manuscript was modified, can be found below. 

While under review, we have had the opportunity to discuss our manuscript and the statistical analyses we conducted with two experts on statistical modeling, Professor Michael Zyphur from the University of Queensland, and Professor Marcello Gallucci from the University Milano-Bicocca in Italy. Based on their expert input, we made a small change in the second and third model as part of our growth curve analyses. Specifically, we included a path from the control variables to the intervention variables. Including this path resulted in a changed effect of professional support on emotional exhaustion: this effect was no longer significant in the final model. All the other effects in the growth curve analyses, such as on work engagement and with organized supportive group meetings, remained. We revised the manuscript accordingly and marked all changes in the revised manuscript. 

We thank you for considering our revised manuscript for publication in PloS One. 

Sincerely, on behalf of all authors, 

Lara Solms

---

## [Decision Letter · Decision Letter 2]

28 Oct 2022

Physician Exhaustion and Work Engagement During the COVID-19 Pandemic: A Longitudinal Survey into the Role of Resources and Support Interventions

PONE-D-22-07749R2

Dear Dr. Solms,

We’re pleased to inform you that your manuscript has been judged scientifically suitable for publication and will be formally accepted for publication once it meets all outstanding technical requirements.

Kind regards,

Juan A López-Rodríguez

Academic Editor

PLOS ONE

Additional Editor Comments (optional):

Reviewers' comments:

Reviewer's Responses to Questions

**Comments to the Author**

1. If the authors have adequately addressed your comments raised in a previous round of review and you feel that this manuscript is now acceptable for publication, you may indicate that here to bypass the “Comments to the Author” section, enter your conflict of interest statement in the “Confidential to Editor” section, and submit your "Accept" recommendation.

Reviewer #1: All comments have been addressed

Reviewer #3: All comments have been addressed

2. Is the manuscript technically sound, and do the data support the conclusions?

Reviewer #1: Yes

Reviewer #3: Yes

3. Has the statistical analysis been performed appropriately and rigorously? 

Reviewer #1: Yes

Reviewer #3: Yes

4. Have the authors made all data underlying the findings in their manuscript fully available?

Reviewer #1: Yes

Reviewer #3: Yes

5. Is the manuscript presented in an intelligible fashion and written in standard English?

Reviewer #1: Yes

Reviewer #3: Yes

6. Review Comments to the Author

Reviewer #1: It is not necessary to make any further corrections. I suggest that the paper be accepted for publication.

Reviewer #3: Thanks for the authors have addressed all comments and the manuscript is now acceptable for publication.

7. PLOS authors have the option to publish the peer review history of their article (what does this mean?). If published, this will include your full peer review and any attached files.

Reviewer #1: No

Reviewer #3: No

---

## [Editor Report · Acceptance letter]

10 Jan 2023

PONE-D-22-07749R2 

Physician Exhaustion and Work Engagement During the COVID-19 Pandemic: A Longitudinal Survey into the Role of Resources and Support Interventions 

Dear Dr. Solms:

I'm pleased to inform you that your manuscript has been deemed suitable for publication in PLOS ONE. Congratulations! Your manuscript is now with our production department. 

Kind regards, 

on behalf of

Dr. Juan A López-Rodríguez 

Academic Editor

PLOS ONE